# The worst graph layout algorithm ever

Sara Di Bartolomeo [iD]*
Northeastern University

Matěj Lang [iD]†
Masaryk University

Cody Dunne [iD]‡
Northeastern University

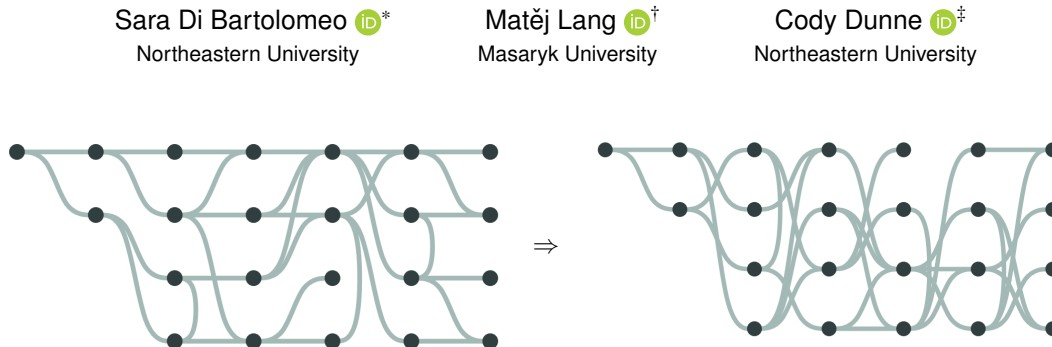

Figure 1: By maximizing edge crossings with our method (WORSTISFIMAL), a conventional layout of a graph with 33 edges and 23 nodes can go from just 4 crossings (left) to an impressive 59 crossings (right). A thrilling and provably-optimally-worst result!

## ABSTRACT

Graph layout algorithms strive to improve the utility of node-link visualizations or graph drawings by optimizing for readability criteria. One such criteria that has been widely used is to count edge crossings. Prior work has focused solely on **minimizing** the number of edge crossings, including provably-optimal layout algorithms for layered graphs. The research community has completely ignored the other side of the coin—can we **optimally maximize edge crossings**? This paper answers this question in the affirmative. Our **WORSTISFIMAL layout algorithm** produces the most **un**readable layered graph drawing. It does so by using linear programming to produce a provably-optimally-awful solution. We hope that this groundbreaking result opens up an entirely new field of inquiry for graph drawing researchers—**optimally-worst layout algorithms**.

## 1 INTRODUCTION

Researchers interested in visualizing graphs have been working endlessly to improve graph drawing algorithms. Graph layout algorithms try to improve the readability of a graph by positioning nodes on the screen so that certain readability metrics are respected, either intentionally or via heuristics that tend to produce good results quickly. Amongst these readability criteria, the most important is to minimize the number of edge crossings [19, 35]. Thus, the holy grail of graph drawing has been to find an algorithm that produces a drawing with the fewest edge crossings—ideally combining this capability with a fast running time [39, 21].

There is, however, an alarming dearth of research on doing the opposite: **maximizing the number of crossings**. Just as it is more challenging to ride a unicycle than a bicycle, we argue that ignoring this second complimentary "wheel" of research undermines the stability of graph drawing as a field. To encourage discussion and widespread dissemination of this potentially-earthshaking limitation (i.e. if the wheeled vehicle of the graph drawing community tips over) we term it **the unicycle fallacy of graph drawing**.

In addition to serving as a call to action for the community, this paper also provides the first groundwork to address the challenge of the unicycle fallacy. We provide a **provably-optimally-worst layout algorithm** for layered graph drawings which maximizes the number of edge crossing, which we term WORSTISFIMAL. Our Integer Linear Programming formulation allows us to use solvers to compute layouts which are guaranteed to have the most edge crossings possible. An example of how bad our results are is illustrated with a small graph in fig. 1. We claim the title of "worst layout algorithm ever" because this layout algorithm is optimal in its awfulness (at least for this readability criteria). The results are literally unbeatable in badness.

The question which arises naturally is: is this layout algorithm useful? The obvious answer is that it is not useful at all—but we believe that further research is necessary in order to verify this hypothesis. However, in the immortal words of GLaDOS, a pioneer in the field of bad outcomes:[1]

> *We do what we must*
> *because we can.*

This paper provides several fundamental contributions:

1. Identifying and raising awareness of the unicycle fallacy of graph drawing, which may affect the fundamental stability of the field.
2. The formulation of the first provably-optimally-worst layered graph layout algorithm for maximizing edge crossings using Integer Linear Programming—WORSTISFIMAL.
3. A free and open-source reference implementation of WORSTISFIMAL as a JavaScript library, which we hope will accelerate research in this area.
4. The design and results of an initial user study to evaluate the efficacy of optimally-worst layout algorithms.

## 2 SUPPLEMENTAL MATERIAL

A copy of this paper along with all supplemental materials is available at https://osf.io/pjctw. We have provided sufficient materials for future researchers and practitioners to reproduce and replicate our results. Including these materials will also provide readers with the baseline code necessary to easily extend our results for creating optimally-worst layouts using other criteria.

## 3 BACKGROUND

The discipline of representing graphs visually is called *graph drawing* [4]. Graphs have been used extensively in all fields of human knowledge: relevant examples include beaucoup convoluted visualizations aimed at exposing the inner workings of neural networks

*e-mail: dibartolomeo.s@northeastern.edu
†e-mail: langm@mail.muni.cz
‡e-mail: c.dunne@northeastern.edu

---

[1] https://half-life.fandom.com/wiki/Still_Alive

[45, 36], several endeavors to get humans interested in SQL queries [15, 28], and the many attempts at explaining the plot of the move The Matrix using graphs[2] [41, 42, 22, 34, 33, 24, 47, 46, 31, 17].

Research on how to best represent a network as a node-link visualization started as early as 1934 with Moreno [29], followed by the first attempts at computational layout approaches in the 1960s [43, 44, 26]. Research on graph layout algorithms has been ongoing since then, producing both general-purpose and specialized layout algorithms, but the problem is still open, and the general agreement is that there is no universal layout solution that embraces all the possible applications and use cases [23, 5].

Our novel approach for computing provably-optimally-worst layouts for layered graphs is based on methods that use Integer Linear Programming [7, 12, 30, 8, 13, 20, 24, 14, 47] to optimize graph readability, especially the methods proposed by Zarate et al. [49] and formulated as a modular library for optimal layered graph drawing by Di Bartolomeo et al. [16]. Although much research has been conducted based on these methods, to the best of our knowledge, none of them have focused on maximizing the number of crossings.

**Readability metrics:** Scientists have developed several readability criteria for visualizations [35, 19], and algorithms to implement those criteria in practice [19, 27]. The number of crossings has been proven to be the metric that most negatively influences the readability of a graph [35, 19], which is why we focus on edge crossings for this fundamental research. Other metrics of interest, but which other researchers have only done in reverse, include maximizing node-node overlaps, maximizing edge length, minimizing angles between crossing edges, and minimizing the angle between all the edges incident to a node.

Although this paper focuses on maximizing the number of edge crossings, lengthy discussions have been going on about **how** to count the number of crossings [32, 37]. We do not wish to have a say in this discussion, and to avoid any confusion we state that we use the **rectilinear crossing number**. This approach counts crossings between edges but considers edges as rectilinear segments, as defined in Pach and Tóth [32]. This is in contrast to the **pairwise crossing number**, which counts every intersection only once, and the **crossing number**, which doesn't have the rectilinear edge restriction. Using the rectilinear crossing number prevents the graph from being able to have infinite crossings, as we allow bends [3].

A discussion about the effect of optimizing the opposite of readability metrics beyond the rectilinear crossing number can be found in section 8.

## 4 METHOD

Linear programming guarantees that the solution is optimal within the allowed boundaries given by the constraints. Previous researchers have developed linear programming solutions for producing drawings with the fewest possible crossings [16, 49]. Our method builds upon these results.

A graph layout algorithm based on linear programming is defined by formulating a problem in terms of an objective function and constraints. The objective function for **minimizing** crossings could look like this example from STRATISFIMAL LAYOUT [16], using the notation from table 1:

$$\text{Minimize} \sum_{k \in L} \sum_{\substack{u_1 w_1, u_2 w_2 \in E_k \\ u_1 w_1 \neq u_2 w_2}} c_{u_1 w_1, u_2 w_2} \tag{1}$$

The objective function can be read as "minimize the sum of crossings for each layer $k$, for each pair of edges $u_1 w_1$ and $u_2 w_2$ in layer $k$". The boolean variable $c_{u_1 w_1, u_2 w_2}$ is equal to 1 if the two edges cross, and to 0 if the two do not cross.

To the best of our knowledge, no research exists on how to produce the **maximum** number of crossings. This is surprising given

---

[2]Inspired by xkcd's beautiful charts: https://xkcd.com/657/

---

**Definitions:**

| | |
|---|---|
| $G = \{N, E\}$ | The graph (network) consists of a set of nodes $N$ and edges $E$. |
| $N_k$ | The nodes in layer $k$. |
| $L = \{1, 2, \ldots, \ell\}$ | The set of $\ell$ layers in $G$. |
| $u_1 w_1$ | An edge between nodes $u_1$ and $w_1$. |

**Decision variables:**

| | |
|---|---|
| $x_{u_1, u_2}$ | The relative vertical order of nodes. Boolean equal to 1 if $u_1$ is above $u_2$, 0 otherwise. |
| $c_{u_1 w_1, u_2 w_2}$ | Indicates if edges $u_1 w_1$ and $u_2 w_2$ cross. Boolean equal to 1 if they cross, 0 otherwise. |

Table 1: The notation used in this paper.

the simplicity of the solution to this problem—though clever solutions often seem simple in hindsight. Indeed, it is enough to just change the first word in the optimization function:

$$\textbf{Maximize} \sum_{k \in L} \sum_{\substack{u_1 w_1, u_2 w_2 \in E_k \\ u_1 w_1 \neq u_2 w_2}} c_{u_1 w_1, u_2 w_2} \tag{2}$$

This simple change will instruct the solver to return a graph layout algorithm with the maximum number of crossings. Now, in order for this to work, we also need to add to the problem formulation a few constraints, which stay exactly the same between minimizing and maximizing crossings. These constraints mostly serve to relate variables such as determining the position of the nodes based on the presence or absence of a crossing. Here we provide an overview of these constraints and their effects. Please see the STRATISFIMAL LAYOUT paper [16] for a more thorough discussion.

The first constraint we need is how we tie the crossing number, $c$, to the relative position of the nodes. The $x$ variables define the relative position of the nodes thus:

$$x_{u_1, u_2} = \begin{cases} 1, & \text{iff } u_1 \text{ is above } u_2 \\ 0 & \text{otherwise.} \end{cases}$$

Imagine two pairs of connected nodes, each pair having one of its nodes on one layer (parallel axis) and the other on a separate layer. Here are all the possible relative positions of these four nodes:

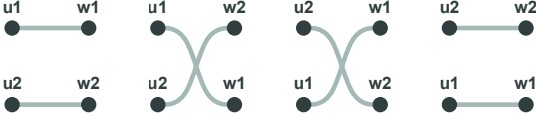

We deduce from this picture that two edges cross when their incident nodes are in inverted positions on the two different layers. More precisely, the two edges cross when (1) $u_1$ is above $u_2$ but $w_1$ is below $w_2$ (second case pictured) OR (2) $u_2$ is above $u_1$ but $w_2$ is below $w_1$ (third case pictured).

Based on the above intuition, we can write a constraint that ties together the values of the $x$ and $c$ variables thus:

$$\begin{aligned} c_{u_1 w_1, u_2 w_2} + x_{u_2, u_1} + x_{w_1, w_2} \geq 1 \\ c_{u_1 w_1, u_2 w_2} + x_{u_1, u_2} + x_{w_2, w_1} \geq 1 \end{aligned} \tag{3}$$

$$(\forall k \in L : \forall u_1 w_1, u_2 w_2 \in E_k^<, \text{ where } u_1 w_1 \neq u_2 w_2)$$

The above constraint will be repeated for every edge in the graph. If we replace the variables with actual values from our illustration

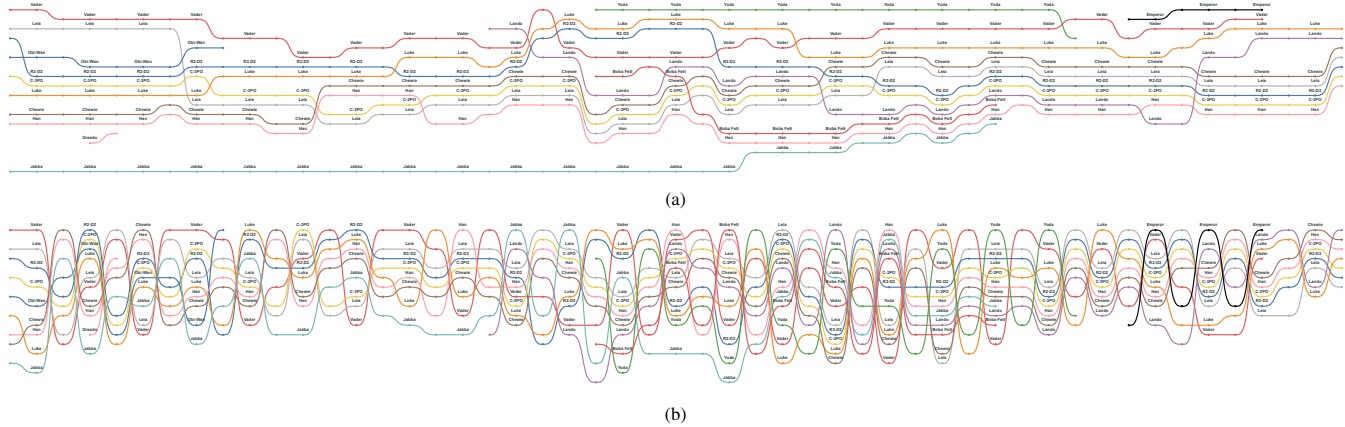

(a)

(b)

Figure 2: A storyline [31] of the original Star Wars trilogy with two layouts, each respecting the constraints on position that storylines impose. E.g., characters that appear in scenes together are drawn nearby vertically in the visualization and scenes are ordered on linear axes from left to right in the order they appear. (a) is laid out to have the fewest possible crossings—37. We created it using the optimal layout algorithm detailed in STRATISFIMAL LAYOUT [16]. (b) instead lays out the storyline to have as many crossings as possible using our WORSTISFIMAL approach. We now have 1953 crossings—a 5178% increase over the state of the art and provably optimally worst!

of relative positions, we will be able to see that $c$ will necessarily turn out to be 1 in configurations in which the relative position of nodes creates a crossing. For example, consider the second case we illustrated above:

$$c_{u_1w_1,u_2w_2} + 0 + 0 \geq 1$$

The value of $c_{u_1w_1,u_2w_2}$, which is restricted to be 1 or 0 by definition (table 1), must be 1 to satisfy this constraint.

The crossings constraint, though, is insufficient to fully capture the problem. We need another constraint to define transitivity in the relative positions of nodes—plainly, the fact that if node $u_1$ is above node $u_2$, and node $u_2$ is above node $u_3$, then node $u_1$ is necessarily above node $u_3$. Writing this as constraints we get:

$$
\begin{aligned}
x_{u_1,u_2} + x_{u_2,u_3} - x_{u_1,u_3} &\geq 0 \\
-x_{u_1,u_2} - x_{u_2,u_3} + x_{u_1,u_3} &\geq -1
\end{aligned}
\tag{4}
$$

$$(\forall k \in L : \forall u_1, u_2, u_3 \in N_k, \text{ where } u_1 \neq u_2 \neq u_3 \neq u_1)$$

These two sets of constraints—the crossing constraint in eq. (3) and the transitivity constraint in eq. (4)—are, together with the optimization function in eq. (2), enough to create a layered graph layout algorithm that optimally maximizes edge crossings.

At this point, we feed the model made of constraints and objective function to an LP solver. The solver will assign values to variables according to the objective function. From the variable values, we can then create the visualization layout, as shown with two examples in figs. 1 and 2.

For solvers, we are particularly fond of Gurobi[3] or glpk.js,[4] the latter of which can run in a browser, but any LP solver can be used for the purpose.

## 5 USEFULNESS

In the field of graph drawing, no justification for the functional utility of solving a problem is often provided or requested. Indeed, solely scientific curiosity about a problem is often enough to justify spending time on it.

---

[3] https://www.gurobi.com/
[4] https://github.com/hgourvest/glpk.js

However, we spent some time trying to come up with cases in which maximizing crossings could be useful. The following subsections will enumerate three different cases in which it might be desirable to maximize the number of crossings.

### 5.1 The best baseline for comparing other algorithms

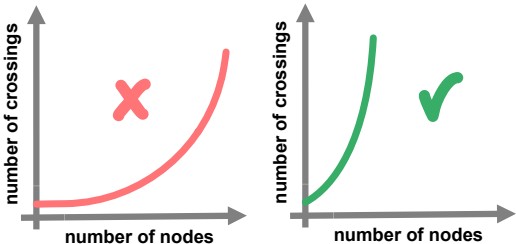

A key feature of our WORSTISFIMAL layout algorithm is that it produces provably-optimally-worst layouts in regards to the number of edge crossings. Thus, it can serve as a benchmark against which other layout algorithms, especially heuristics, can be compared. Between our method and STRATISFIMAL LAYOUT [16], which conversely produces provably-optimally-best layouts, we now have concrete upper and lower bounds with which to evaluate the efficacy of a layered graph layout algorithm. Naturally, any researcher searching for a straw man against which to compare their approach should use WORSTISFIMAL, which is virtually guaranteed to be worse than even a random layout!

### 5.2 Obfuscating circuits

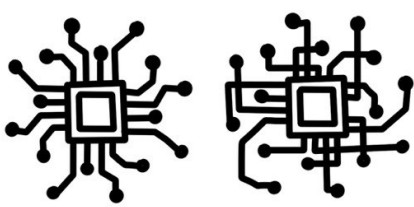

The layout of electronic circuits has been a field of study adjacent to graph drawing. In circuit design, connected elements (circuit components) must be positioned on a plane (the circuit board) while

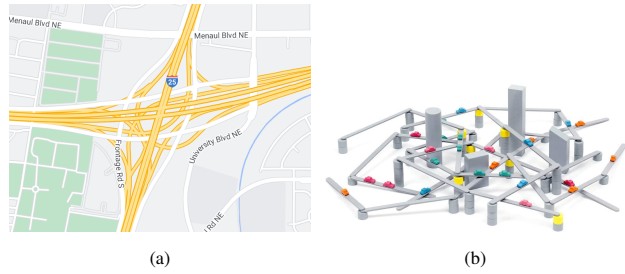

(a)             (b)

Figure 3: Two examples of seemingly complex-by-design highway interchanges: (a) The Big-I interchange in Texas, which is a five-level interchange, as it appears on Google Maps. (b) The board game Tokyo Highway, in which the goal is maximizing intersections between highways.

following strict criteria. Crossings within a plane could create short circuits. The problem has been explored by Sugiyama [40], included in a survey of graph layout problems [18], and explored by many representatives of graph drawing [1]—all with the goal of making the graph more readable.

However, because a good layout might aid in the readability of the circuit, it can also lead to the electronic devices being more susceptible to reverse engineering, modifications, and industrial espionage. The area of circuit obfuscation focuses on deterring such activities by making it difficult to understand the circuit design [48]. Provably-optimally-worst layout algorithms can play a role in creating such obfuscated designs.

### 5.3 Designing complex highway interchanges

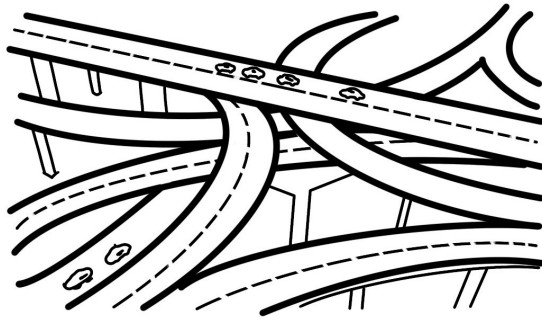

Designers of highway interchanges seem particularly fond of crossings, at least for when their designs are viewed from above. It is almost as if they derive some perverse pleasure from creating intertwined and convoluted structures that confuse unaware drivers just trying to get from A to B. A real-world example of this phenomenon can be seen in fig. 3(a), showing a five-level interchange in Texas. There have even been board games created that indoctrinate interchange design hobbyists into these antisocial schools of design thinking (fig. 3(b)).

### 6 USER STUDY

To validate that the provably-optimally-worst layouts created by WORSTISFIMAL are actually bad from a user perspective, we conducted a rigorous user study. We recruited 3 participants for our experiment. We asked our participants (P1, P2, P3) to comment on WORSTISFIMAL visualizations and offer their feedback. P1 commented: "These visualizations are horrible." Likewise, P2 said: "I am an expert at reading graphs, and I hate these." P3 went even further, stating: "I don't even see the point."

Overall, based on the results of our extensive investigation, we deem that these comments reflect positively on our intentions when conducting this research.

### 7 DISCUSSION

Note: The entire contents of paper is a joke intended to raise awareness of optimal layout algorithms, graph drawing readability criteria, and curious academic conventions. The WORSTISFIMAL technique described in section 4 does work and did generate the visualizations in the paper, but we did not actually run a user study. Several of our statements throughout the paper, esp. regarding the motivations of others, should be supplemented with a large chunk of NaCl.

### 8 FUTURE WORK: MORE METRICS TO OPTIMIZE

This paper provides the first formulation for a layered graph layout algorithm that maximizes edge crossings, creating the first spokes in the second wheel graph drawing needs to address the unicycle fallacy. However, there is much more to be done to fill out that wheel and bring stability to our field.

To guide future researchers in populating these spokes, we now provide a set of metrics that we believe will be most productive for the community to address. Some of these are based on research that has been previously done regarding readability criteria [35, 19].

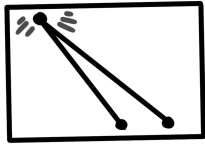

**Maximizing edge length:** Another classic metric for graph layout algorithms is minimizing edge length. However, while the worst number of crossings is a finite number, edge length is not—the worst value for edge length would be infinite, making the graph difficult to visualize on a finite plane. We leave this challenge for future research.

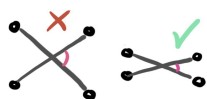

**Hard to read angles:** In general, in a non-planar graph, it is advisable to position nodes so that edges that cross form ample angles— preferably 70°according to Huang et al. [25]. Ample angles help users to follow the intended path after a crossing. Optimizing for acute angles, or, even better, overlapping edges will improve the unreadability of the result.

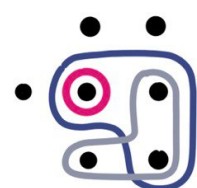

**Group overlap maximization:** In some instances, graphs might have groups of nodes that should be kept adjacent [38, 11]. Avoiding overlaps between groups of nodes is often a constraint in layout algorithms. Keeping the groups as distinct as possible helps greatly in understanding group membership. Conversely, in order to disrupt the readability of a visualization, we could try to draw every group such that its bounds or its membership overlaps with as many other groups as possible.

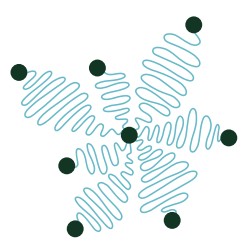

**Maximizing edge bendiness:** Some layout techniques allow for edges to have bends. While bends can be useful to avoid crossings, they reduce the readability of a graph. Steps for reducing edge bendiness are indeed a common inclusion in graph layout algorithms [16]. We wonder what a graph would look like if we instead maximized the bendiness of edges.

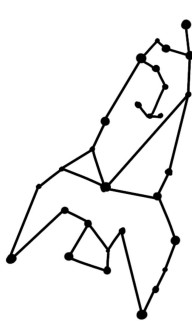

**Rocketshipness**: We took inspiration from Alberto Cairo's Datasaurus [9], in which he extends Anscombe's argument [2, 10, 6] that data points with the same summary statistics can have many different shapes, thus it's important to plot them properly to understand the distribution. Alberto Cairo reinforces this argument by drawing a dinosaur with the same summary statistics. We consider a similar idea by optimizing the graph to resemble—as much as possible—the shape of a rocketship while maintaining the same summary statistics. While Anscombe's summary statistics for a scatterplot are X and Y average position and standard deviation, summary statistics for a graph layout can be, for example, the number of crossings, edge length, or aspect ratio.

Another key challenge for future researchers is to extend our WORSTISFIMAL layout algorithm to work on other types of graphs and more general layouts. It is important to note that our current formulation of the problem only works on layered graphs. Extending it to non-layered graphs would open up plenty of incredible possibilities for making highly-unreadable graph layouts.

## 9 CONCLUSION

This paper details and raises awareness of the unicycle fallacy of graph drawing which may affect the fundamental stability of our field. We also provide the first provably-optimally-worst layout algorithm for layered graphs that maximizes edge crossings. Our use of Integer Linear Programming guarantees optimality, but also long run times and high usage of computational resources. We discuss the utility of this approach, both theoretically and from a user perspective, and how it can be extended by future researchers to fill in the spokes of the second wheel of the graph drawing bicycle.

### ACKNOWLEDGMENTS

We thank Jane Adams for randomly coming up with this idea during a meeting and Michael Davinroy for providing witty comments about the results.

The work in this paper is inspired by our previous research on graph layout algorithms [16, 17] which was supported by NSF grant IIS-2145382.

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
