# OpenReview forum: "The worst graph layout algorithm ever"
_IEEE.org/2022/Workshop/altVIS — Accept_

### Official Review · Reviewer_mhPz · 2022-08-05

**Review:**

This makes a few thought-provoking points, and is definitely a "weird" submission that wouldn't make it to a serious conference (though with some adjustment it could be a SIGBOVIK submission).

However, I have quibbles with some of the things that it says.

## Additional related work

The paper focuses exclusively on graph drawing; the future work section could mention that Quality Metrics have been used in other areas of visualization [0], opening the door to producing pessimal versions of other types of visualization.

Michael Correll's "Ross-Chernoff Glyphs" paper [1] is relevant as an previous example of introducing a deliberately bad visualization (the Ross-Chernoff Glyphs) to provoke thought, and for discussing why a bad idea (Chernoff faces) might be widely circulated in academia.

Outside of visualization, pessimal algorithms have also been investigated in the context of search (e.g., [3], [4]); however, this work focused on pessimal running-time, rather than pessimal result quality, so this work is in some sense dual in its perversity.


## Specific points

The title of this paper is unjustifiably broad: as the background section acknowledges, the presented algorithm is only pessimal in the number of edge crossings. An alternative algorithm that assigns all nodes to the same position is superior in both computational complexity and unread-ability of results (a slight refinement of this algorithm would assign everything to the same position, at a point outside the visible region).

The statement "several endeavors to get humans interested in SQL queries [15, 28]" is misleading, as reference 15 is not about SQL, but rather SPARQL (a very different query language that is applied to RDF triples rather than tabular data in a relational database)

Rather than changing the first word in the optimization function from "minimize" to "maximize", it would be more conventional to add a minus sign; this is obviously equivalent, but keeps the optimization problem in standard form. However, the approach currently taken by the paper is not wrong.

Much of the "Method" section (from "Please see the STRATISFIMAL LAYOUT paper [16] for a more thorough discussion" onwards) could be deleted. On the one hand, it’s good for papers to be self-contained and understandable in isolation. On the other, the extra content extends the paper, is not novel, and isn’t important to the key point that one can take an optimal method and change the sign to do pessimization instead.

In the Method section, there are two uses of "LP solver" that should instead be "ILP solver": a general Linear Programing solver cannot necessarily handle the integer constraints in an Integer Linear Program.

An additional potential application is to intimidate or make things seem more complicated than they really are as a rhetorical trick.

The Note in the discussion is clunky: explaining that something is a joke is usually not funny. Personally I would instead re-phrase section 6 to something like:

> We considered running a rigorous user study to be needlessly cruel. However, were we to run such a study, we would expect to received participant feedback such as "These visualizations are horrible", "I am an expert at reading graphs, and I hate these", and ""I don’t even see the point".

The answer to "what a graph would look like if we instead maximized the bendiness of edges"  is perhaps "like Absyss Explorer" [5].

The "Rocketshipness" example is in some ways similar to TSP art [6], which aims to approximate an image by stippling, and then joining the stippled points with a single curve (by solving a Traveling Salesman Problem).  This essentially approximates an image by laying out a network with a specific topology (a linear chain); the Rocketshipness example approximates an image by laying out a network whose topology is determined by the input data.

Some things that should be capitlized in the references are not (e.g., "sparql" and "htn")


## Minor edits

-  There are many uses of "fig" that should be capitalized (e.g., "fig. 1" -> "Fig. 1")
- "optimally worst" -> "pessimal"
- "beaucoup" - perhaps italicise as a French loanword?

[0]: https://onlinelibrary.wiley.com/doi/10.1111/cgf.13446

[1]: https://research.tableau.com/sites/default/files/altCHI-preprint.pdf

[2]: http://ieeexplore.ieee.org/document/5290690/

[3]: http://www.hermann-gruber.com/pdf/fun07-final.pdf

[4]: https://dl.acm.org/doi/10.1145/990534.990536

[5]: https://ieeexplore.ieee.org/document/5290690

[6]: https://wiki.evilmadscientist.com/TSP_art


**Conflicts:**

None

**Review Inclusion:**

Yes

**Sufficiently Alt:**

Yes

**Superlative:**

Worst graph layouts

---

### Official Review · Reviewer_zK5X · 2022-08-25

**Review:**

Great paper, very “alt.”

I would challenge two points here.

The first is on the proclaimed uselessness of the algorithm.

I don’t think this is necessary true. Beyond the potential applications mentioned where edge crossings are potentially desirable, the exercise itself has utility, by making the phenomena of “graph legibility” what Heidegger calls “present at hand” (well, “Vorhanden”, but you get the idea): you don’t learn much about a machine when it is operating, but when it breaks you are forced by necessity to examine its component parts and how it operates. This algorithm performs the same procedure. Look at the future metrics section, for instance: all of these components are things that come up as being things that obfuscate graph legibility, but how many graph drawing algorithms take them into consideration when performing their optimization? In the disfunction we learn more about the function. So a useful exercise even if the algorithm is not useful, since it forces us to take seriously the desiderata of graph drawing algorithms.

The second point is I feel the potential set of metrics is perhaps a little under-ambitious.

Once we are in the realm of interactive graphs, then we’ve got plenty more metrics to really hide things. For instance, overplotting, or sending points randomly into the far reaches of the canvas (and so smushing all the other points close together), or even just reshuffling the location of points from frame to frame (I might even relax some of the other constraints if the points were literally exhausting to re-locate every time something changed).

Other issues:
The fake user study I think is a bit unnecessary, and it’s a bit misleading if there were, in fact, no P1s, P2s, or P3s. If you need the quotes, feel free to attribute them to me. I will go on record to say:
“These visualizations are horrible”, “I am an expert at reading graphs, and I hate these.” and “I don’t even see the point” even though the second statement is a lie, and the first statement directly contradicts the first (and arguably most central) point of my review.

**Conflicts:**

No known conflicts

**Review Inclusion:**

Yes

**Sufficiently Alt:**

Yes

**Superlative:**

The worst (algorithm)

---

### Official Review · Reviewer_tfkB · 2022-08-25

**Review:**

The research is ground-breaking in controversially studying the effect of optimizing the opposite of readability metrics beyond the rectilinear crossing number.
Pros:
1. the research idea to challenge the pre-assumed readability metrics is original and courageous
2. the authors provide supplementary materials that help future research: https://osf.io/pjctw
3. the study makes alerts us the to improve the diversity in readability metrics
Cons:
1. Section 7 discussion is lack of contents and references. Maybe the authors can integrate Section 7 into Section 8.
2. The authors can improve Section 6 by providing more background of the participants and recruiting methods to reason for the conditions that the positive comments apply

**Conflicts:**

NA

**Review Inclusion:**

Yes

**Sufficiently Alt:**

Yes

**Superlative:**

Most contraversial

---

### Official Review · Reviewer_gbxK · 2022-08-30

**Review:**

Meta review:

Absolutely the worst; strong accept. In all seriousness: reviewers generally found the paper to be a good fit for alt.VIS and of sufficient contribution. Figures are particularly wonderful, and future work is extremely thoughtful. Several reviewers noted that the "user study" section fell flat in humor and would benefit from some tonal adjustments. Regarding another review comment that "An additional potential application is to intimidate or make things seem more complicated than they really are as a rhetorical trick", you may appreciate "Towards a Theory of Bullshit Visualization" by Michael Correll from a past year of alt.VIS -- in particular, his reference of DefenseCharts (T. Hwang. Defense charts. https://twitter.com/ DefenseCharts, 2021.), which catalogues particularly egregious visualizations from the Department of Defense, and feature a number of graph layouts that might very well be approaching maximal complexity. Authors should note suggestions for improvements to flow and format in the reviews.

**Conflicts:**

I am in the same lab as two of the authors, and listed in the acknowledgements section.

**Review Inclusion:**

Yes

**Sufficiently Alt:**

Yes

**Superlative:**

The Worst Algorithm at alt.VIS 2022

---

### Decision · Program_Chairs · 2022-08-31

Accept